

# Visual design of green information in urban environment based on global similarity calculation and multi-dimensional visualization technology

Junru Wang

Zhengzhou Vocational University of Information and Technology, Zhengzhou, China

## ABSTRACT

In recent years, the escalating prevalence of elevated consumption and carbon emissions within urban operations has reached a disconcerting extent. This surge in resource depletion and environmental pollution exerts an adverse influence on the well-being of individuals, while impeding societal progress and hindering the enhancement of overall quality of life. Within the domain of urban environmental design, the integration of visual displays emerges as a superior approach to facilitate the assimilation and analysis of green and low-carbon information. However, urban environmental data usually contains multiple dimensions, so it is a problem to realize the data representation of multiple dimensions while maintaining the correlation and interactivity between data. To surmount the challenge of visualizing such intricate information, this investigation initially employs a sophisticated memory-based clustering algorithm for information extraction, accompanied by a global similarity algorithm that meticulously computes attribute component quantities within specific dimensions of the vector. Furthermore, leveraging the inherent power of Vue's bidirectional data binding capabilities, the study adopts the esteemed MVVM (Model-View-View-Model) pattern, fostering seamless two-way interaction through the established logical relationship. As a result, the amalgamation of multidimensional visualization technology empowers comprehensive data mining through a captivating visual augmentation. Concurrently, the application of data visualization dimension control delivers tailored displays tailored to green and low-carbon scenarios within urban environmental design. Experimental results impeccably validate the effectiveness of the proposed algorithm, substantiated by a mere 1.77% false alarm rate for data stream difference detection and a clustering difference of 1.34%. The aforementioned algorithm accentuates the efficacy of visual displays, thus engendering a profound synergy between the industrial and supply chains. Moreover, it facilitates the design, production, and utilization of environmentally friendly products and energy sources. This, in turn, serves as a catalyst, propelling the widescale adoption of green and low-carbon practices throughout the entire industrial chain, fueled by the seamless integration of multimedia data.

Corresponding author
Junru Wang, hgd371@163.com

## INTRODUCTION

Multimedia, with its versatile elements of graphics, text, sound, and moving images, creates an optimal environment for education, graphic design, and various other fields. In particular, the emergence and intervention of multimedia technology in today's urban environmental design have fundamentally transformed and developed the entire environmental design industry. However, as the world continues to focus on achieving better urban environmental development in a low-carbon economy, visualizing green and low-carbon information in urban environmental design becomes crucial in enabling relevant departments to analyze and design effective strategies.

To this end, environmental accounting serves as a means to determine, calculate, record, and report enterprises' activities in environmental development, pollution, and management, ultimately measuring their green and low-carbon situation to visualize the necessary information (*Pan & Yue, 2022*). Yet, in the context of multimedia, data can manifest in various forms, including graphs, texts, audio, and videos, posing significant challenges to environmental accounting's ability to capture each industry's green and low-carbon situation quickly and accurately. In China, as the world's largest emitter, the magnitude and high-dimensional features of green and low-carbon data make it even more difficult to analyze effectively, resulting in a significant reduction in the accuracy of green and low-carbon data information processing due to the lack of effective analysis methods. The increasing amount of data and the inability to interpret them effectively creates a pressing need for technology capable of processing and interpreting vast data, and visualization is created in response to this requirement.

With the progression of social informatization and the increasingly ubiquitous utilization of the Internet, the *corpus* of information sources is growing in size. Apart from the demand for storing, transmitting, retrieving, and classifying the vast amount of green and low-carbon data for better urban environmental design, there exists an urgent need to comprehend the interrelationships and developmental tendencies among the existing green and low-carbon data for better environmental protection strategies. Thus, numerous crucial information and patterns remain obscure within this burgeoning data, and individuals strive to analyze them at a more sophisticated level to make the most of these green and low-carbon data. Whereas data mining is a process that involves extracting information from large amounts of data, its heavily automated algorithmic nature poses a challenge for users to apprehend, explore, and optimize the data and algorithmic process itself. The burgeoning field of visualization, driven by multimedia data, has resulted in a significant amount of work aimed at discovering how visual methods can facilitate the data mining process and render the data more intuitive for users to comprehend and investigate. This necessity has stimulated the development of data mining techniques (*Yang et al., 2020*).

Meanwhile, in scientific research and daily work, individuals are frequently confronted with massive amounts of high-dimensional data that comprise extremely rich and similar objective information. Representing these massive data visually or obtaining concealed information and patterns that interest users has always been an ongoing objective of

scholars. Visualization techniques (*Burch & Timmermans, 2020*) can convert data information into intuitive, graphic, or pictorial representations of physical phenomena or physical quantities that vary in time and space, enabling users to observe things or phenomena that are not visible in the traditional sense. Research in visualization is expanding rapidly. Document (*Zheng et al., 2006*) is based on the combination of Web, 2DVR, 3DVR and GIS technology to develop a virtual city system and realize city visualization. Literature (*Zhang et al., 2021*) explores the dynamic process of management decision from three aspects: image display, image understanding and image application, proposes the auxiliary decision model of big data visualization, and discusses the application system of urban visual governance based on it. All these studies only consider the use of image technology to realize visualization, and do not conduct in-depth discussion on environmental low-carbon issues in cities. Therefore, it holds far-reaching significance to combine data mining technology and visualization technology to visualize green and low-carbon information in urban environmental design. This integration can improve the ability of high-dimensional data mining and data analysis and processing in the multimedia context and lay the foundation for decision-making.

## RELATED WORKS

Data mining is a novel science aimed at extracting useful information from massive data in the information society. It is a technique for processing large data that combines traditional statistics, artificial intelligence (*Hasan et al., 2023*), pattern recognition (*Jain, Duin & Mao, 2000*), database and other fields, and its amalgamation of traditional analytical methods and complex statistical algorithms is increasingly prevalent. According to literature (*Chen et al., 2021*), the data mining process begins with identifying the research object in the face of massive data. A well-defined objective can determine the corresponding data processing models and methods. In the second stage, the data set from the database is selected based on the parameter indices required for the analysis model, and the selected data is pre-processed. The literature (*Song & Wu, 2022*) builds, evaluates, and analyzes the analysis model based on the selected data mining algorithm as per the research needs. Finally, the research object is improved and optimized based on the data analysis results. Artificial neural network algorithms (*Wu & Feng, 2018*), cluster analysis (*Cheng et al., 2022*), and decision tree analysis (*Jun Lee & Siau, 2001*) are the primary methods involved in data mining techniques at this stage.

As data volume rapidly expands, the amount of data processed in data visualization has also increased significantly. Generating images using visualization algorithms is relatively complex and has historically required the use of powerful computers and high-end graphics workstations. However, recent advancements in PC capabilities, various graphics cards, and the development of visualization software have expanded visualization technology to various fields, including scientific research, environmental design, military, finance, and biology. Data visualization is the most intuitive method for job seekers and developers to understand key features in explicit (*Jawaheer, Weller & Kostkova, 2014*) and implicit feedback (*Aljunid & Huchaiah, 2022*). Currently, visualization and analysis tools and methods are becoming more mature and convenient. Applications such as Tableau

and PowerBI can be quickly launched, and tool libraries such as matplotlib, seaborn, echarts, pyecharts, *etc.*, can be used for feature customization development.

Currently, a multitude of scholars have made remarkable contributions to the fields of data mining and visualization, respectively. Consequently, researchers have begun to delve into classification strategies that merge data mining and visualization techniques (*Cortez & Embrechts, 2013*; *O'Halloran et al., 2018*; *Atitallah, Driss & Almomani, 2022*; *Masood et al., 2023*). In particular, the literature (*Fayyad, Grinstein & Wierse, 2002*) presents methodological constructs for knowledge discovery and extraction, in conjunction with visualization, to aid in data mining displays. Meanwhile, the literature (*Shao et al., 2022*) centers on visualization methods that leverage data mining techniques to enhance the visual mapping and data processing stages of visualization. These two approaches each possess distinct tasks throughout the process of transforming raw data into knowledge representations, making it possible to apply data mining methods to generate two-dimensional projections of data points during the data-to-visualization transformation. Additionally, the literature (*Mehbodniya et al., 2022*) introduces a classification scheme that incorporates the degree of visualization and data mining involvement as classification criteria, effectively integrating visualization and data mining techniques to optimize the presentation of hidden data information. Lastly, the literature (*Romero & Ventura, 2020*) puts forth a classification strategy based on the types of user involvement, which describes the user-based visualization exploration participation in data mining algorithms, consisting of information flow entities and target entities.

## GREEN LOW-CARBON INFORMATION VISUALIZATION BASED ON URBAN ENVIRONMENT DESIGN

The purpose of data visualization techniques is to facilitate users in uncovering pertinent information from vast amounts of data and to mitigate the impact of objective factors on the data. Ideally, visualization techniques enable users to draw insightful inferences by simply viewing the data. Given this potential, data visualization has widespread applications in data mining and exploration, information retrieval, strategic analysis, and business intelligence (*Moradi et al., 2018*). As depicted in Fig. 1, this article designs the data visualization process by taking into account certain psychological approaches and then utilizing computers to simulate human cognitive systems to present the data.

To ensure accurate and efficient comprehension of the data while avoiding any distortion of information, visualization techniques must be congruent with the human perceptual system. However, the challenge lies in the fact that the more abstract perceptual processes, which are not directly related to cognitive processes, must be considered when developing data visualization systems. To make the analysis and manipulation of data more effective, the structure of the information conveyed by the system must be compatible with the requirements of data representation and the preferences of human cognitive processes. Subsequently, this article visualizes the implementation of green low-carbon information by extracting knowledge from low-carbon data in urban environments and then conducting data mining and analysis of the acquired knowledge to actualize urban design amidst data diversity.

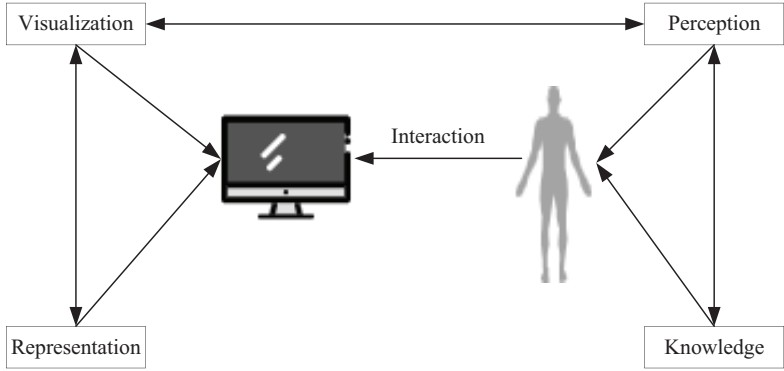

**Figure 1 Visualization process.**

## Knowledge extraction framework

To visualize green and low-carbon information in urban environmental design, this article integrates data mining and visualization technologies to extract knowledge from low-carbon information on the network. Figure 2 illustrates the entire process, which uses keywords such as low-carbon, carbon neutral, carbon cycle, and footprint and employs automatic data mining algorithms to develop corresponding models based on existing data. Through interpretation and validation of the models, analysis results or embedded knowledge is extracted. Figure 3 depicts visual mapping based on this knowledge, using visualization techniques to represent the input data in a certain visual form. The user can explore and extract features and patterns that meet their requirements from the visualization results *via* visual channels. In this interactive visual model, the extracted knowledge results generate new visualization outcomes by receiving interactive feedback from the user.

## Similarity-based algorithm for arranging green low-carbon information dimensions

Cluster analysis, a statistical technique that has been extensively researched for many years, has mainly focused on improving distance-based cluster analysis, while cluster analysis tools based on K-means, K-medoids, and other methods have been integrated into statistical analysis packages or systems. In visualization implementations, observational learning using cluster analysis can eliminate the influence of example-based learning. In contrast to traditional clustering, which uses geometric distance to measure similarity, concept clustering groups objects together only if they can be described by a common concept. Cluster analysis can be applied to perform untrained observational learning on data when used in data mining for digital visualization. This article uses a memory-based clustering algorithm to mine information from a dataset of low-carbon information containing n data objects. The K-means algorithm is used to randomly select K initial cluster centers from known data points, and then iteratively assign each data point to the nearest cluster center, and update the location of the cluster center until the convergence condition is met. During this iteration, all data points are stored in memory. A similarity matrix is used to store the proximity between two of the n objects. First, for each number of

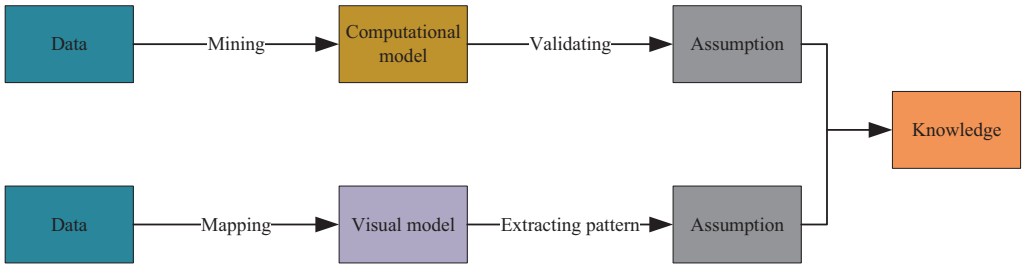

**Figure 2 Data mining and information visualization.**

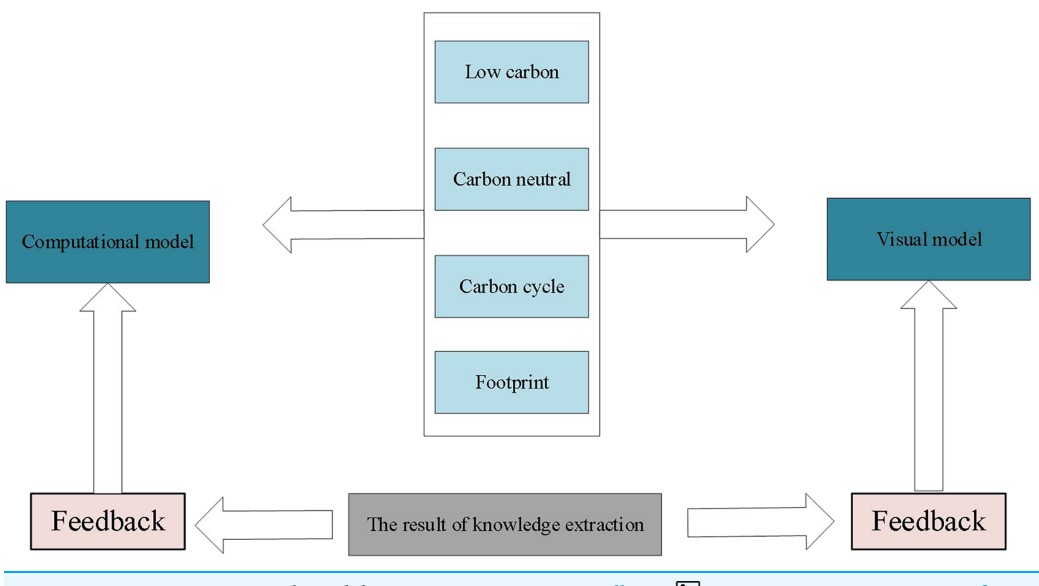

**Figure 3 Interactive visual model.**

points, calculate the similarity between them. By calculating the similarity between all data points, a similarity matrix is constructed. Each element of the matrix represents a similarity score between two data points. On the basis of the similarity matrix, the logarithmic data points are sorted using a suitable ranking algorithm. Finally, the optimal sorting result is obtained by adjusting the parameters based on the clustering algorithm. Let the similarity matrix be as follows:

$$
\begin{bmatrix}
0 & & & \\
d(2,1) & 0 & & \\
\vdots & \vdots & 0 & \\
d(n,1) & d(n,2) & \dots & 0
\end{bmatrix}
\tag{1}
$$

Here $d(i,j)$ is a quantitative representation of the similarity between $d_i(x_{i1}, x_{i2}, \ldots, x_{in})$ and $d_j(x_{j1}, x_{j2}, \ldots, x_{jn})$. Since $d(i,j)$ and $d(j,i)$ represent the same meaning, only the triangular array is written down, and the value of $d(i,j)$ is larger when objects i and j are more similar to each other and closer to 0, and *vice versa*. By calculating the similarity of data, it is possible to differentiate, classify and cluster the analysis of massive data. The

simplest way to calculate the similarity (or difference) between objects is based on the distance between objects. The distance function is as follows:

$$d(i,j) = \sqrt[q]{w_1(x_{i1} - x_{j1})^q + w_2(x_{i2} - x_{j2})^q + \ldots + w_n(x_{in} - x_{jn})^q} \tag{2}$$

where the values of $w_i (i \in \{1, 2, \ldots, n\})$ and $q$ are taken as follows

$$\begin{cases} w_i = 1, q = 1 \\ w_i = 1, q = 2 \\ w_i = 1, q \geq 2 \\ w_i \in \mathbb{Z}, q = 2 \end{cases} \tag{3}$$

After obtaining the similarity matrix between data objects, a similarity-based method can be employed to compare the similarity of each dimension in the parallel coordinate system. This will address the issue of data organization affecting the display effectiveness. An optimization algorithm can then be utilized to derive the best combination scheme for guiding the visualization display. The similarity between dimensions is obtained as follows:

$$V_i = (x_{1i}, x_{2i}, \ldots, x_{ni})^T \tag{4}$$

The global similarity algorithm takes all the data as the object of study and computes a vector of attribute components of all the numbers choked in a certain dimension. In this way the laws implied within all the data can be represented quantitatively, let

$$mean(V_i) = \frac{1}{N} \sum_{j=0}^{N-1} a_{ij}, \text{ then we get:}$$

$$S(V_k, V_i) = \sqrt{\sum_{i=0}^{N-1} ((a_{ik} - mean(V_k)) - ((a_{il} - mean(V_l)))^2} \tag{5}$$

This concludes the dimensional arrangement of the multidimensional data and enables clustering of the data for urban environment design.

## Multi-dimensional information visualization

The visualization of green and low-carbon information is carried out in three main steps. Firstly, the information is collected from open platforms and networks using keywords related to low-carbon information. Secondly, the collected information is processed in a knowledge extraction framework, where data mining techniques are used to extract relevant information and present it in a simple and attractive form of knowledge representation, while discarding irrelevant information. Thirdly, the extracted multidimensional knowledge is arranged in dimensions, and the global similarity of images is determined to obtain the structure of the training model. After unsupervised learning of the training set, the design is evaluated based on the effectiveness of green low-carbon information visualization design criteria (as shown in Fig. 4). The design is implemented using the MVVM (Model-View-View-Model) pattern, utilizing the feature

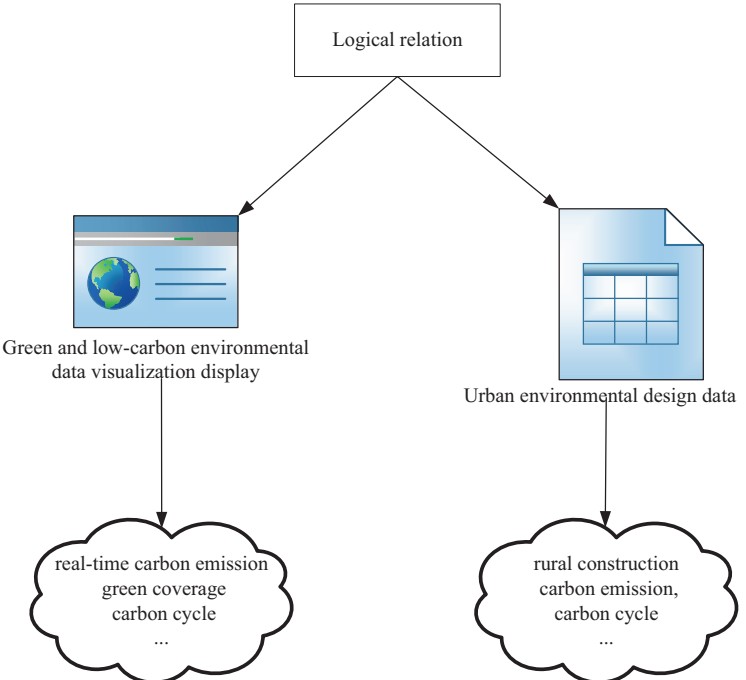

**Figure 4** **MVVM pattern visualization design.**

of two-way binding of Vue data, and enabling two-way interaction through the agreed logical relationship between View-Model. We set up a bidirectional data binding mechanism between the View and the view-model. This means that when the View's data changes, the View-Model updates accordingly, and when the View-Model's data changes, the View updates accordingly. When a user interacts with a View, the view passes the results of that interaction to the View-Model through a data binding mechanism. These interactions can be captured and processed using command patterns or other event handling mechanisms. When the View-Model's data changes, it passes those changes to the View through a data binding mechanism. The view senses these changes and updates its display accordingly.

## EXPERIMENT AND ANALYSIS

In order to evaluate the proposed algorithm, this article compares it with three previous studies: literature (*Song & Wu, 2022*), literature (*Shao et al., 2022*), and literature (*Mehbodniya et al., 2022*). The experimental data, environment, results, and model analysis are presented in the following sections. Literature (*Song & Wu, 2022*) focused on implementing selected data mining algorithms for visualization, while literature (*Shao et al., 2022*) focused on visualization as a form of data mining. Literature (*Mehbodniya et al., 2022*) synthesized the previous two studies and used the degree of involvement of visualization and data mining as classification criteria. Compared to these studies, the algorithm proposed in this article improves in both correctness and efficiency by combining data mining and visualization.

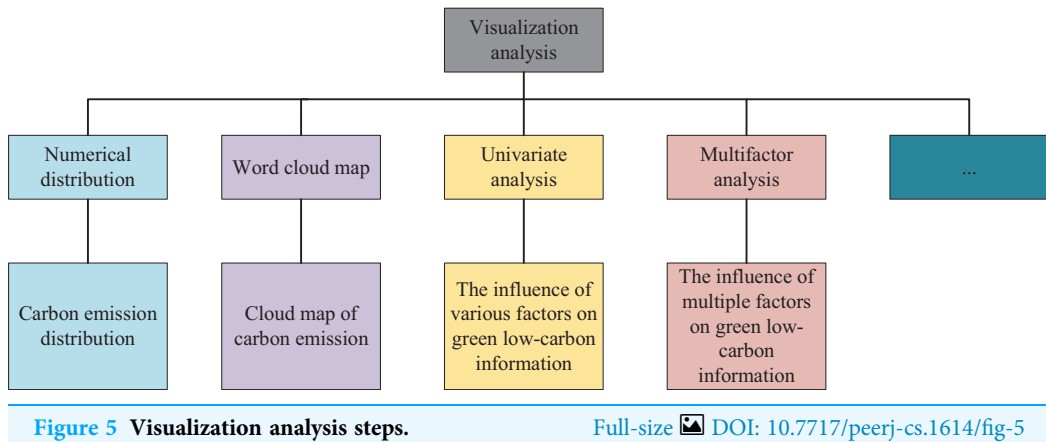

**Figure 5  Visualization analysis steps.**     

## Data processing

For the experimental data utilized in this article, one portion was acquired through Python, while the other was obtained *via* DataFountain (DF platform), a preeminent collaborative innovation platform for data science and artificial intelligence in China, which furnishes artificial intelligence datasets, Big Data competitions, and open source sharing communities. An experimental dataset was procured from the DataFountain platform for the purposes of this article, and the efficacy of the proposed scheme was substantiated using this dataset. In addition, feature cleaning was executed on the dataset to ameliorate the effectiveness of the data. Details pertaining to the data cleaning process encompassed missing value processing (*Rahman & Islam, 2011*), outlier processing (*Dong & Pan, 2021*), normalization processing, Top-N category extraction (*Wu, Macdonald & Ounis, 2020*), and category merging. Carbon emission is an indispensable indicator in urban environmental design and low carbon analysis. The count of missing values in carbon emissions is generally trifling, and a definite value or interval of values is assigned to most industries. As evinced in Fig. 5, among the 8,720 records collected, merely 105 records in the carbon emissions field had missing values, which can be effortlessly expunged for analysis purposes owing to their scant quantity. Subsequently, the upper limit value of carbon emission, lower limit value of carbon emission, average carbon emission, and logarithmic average carbon emission were calculated and recorded in four novel columns, respectively. After once more cleaning the data features, the interrelationships between each feature in the dataset were clarified, followed by the selection of an appropriate graph for visual representation. The visualization was executed through multiple dimensions such as numerical distribution observation, single-factor analysis, multi-factor analysis, and word cloud diagram.

## Comparative performance

As illustrated in Fig. 6, the comparative analysis of green low carbon information for clustering was conducted, wherein the results of literature (*Song & Wu, 2022*), literature (*Shao et al., 2022*), and literature (*Mehbodniya et al., 2022*) were contrasted. Ultimately, the cluster analysis dataflow anomaly detection group proposed in this article demonstrated a

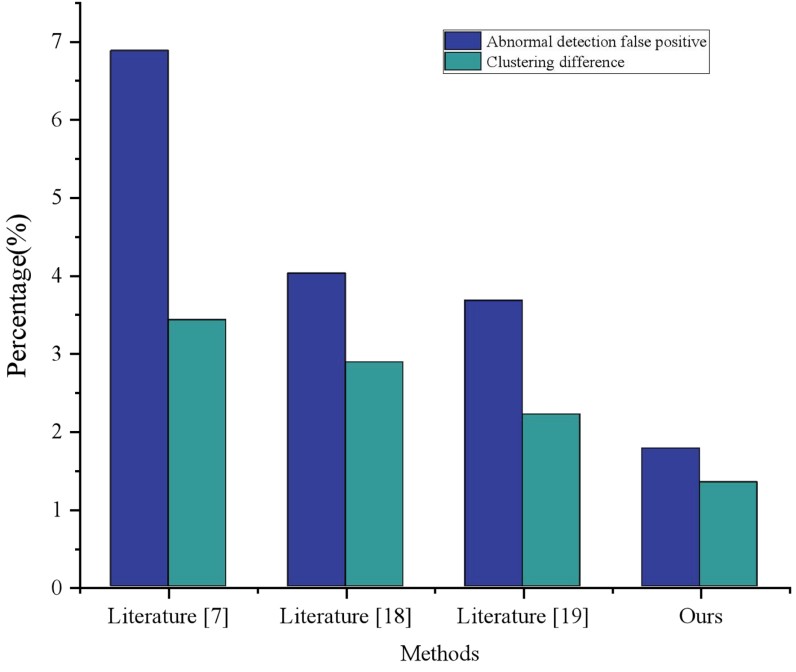

**Figure 6 Comparison of clustering performance with other methods.**

dataflow anomaly detection false alarm rate of merely 1.77% and a clustering difference of 1.34%, thereby realizing a 56% performance enhancement over literature (*Shao et al., 2022*), as well as a 46% performance improvement overall. As evidenced by its clustering effect, the model advanced in this article achieved the most exceptional performance with regard to the expression of urban green and low carbon information visualization.

Moreover, the stability of the interconnected relationship between relevant low-carbon information serves as a measure of information's diversified display capability. Typically, the larger the result of the interconnection coefficient calculation, the stronger the diversified display capability of information, thereby enhancing the practicality of the platform system. Thus, an evaluation experiment is conducted to measure the diversified display capability, and the experimental results are presented in Fig. 7. Given that literature (*Mehbodniya et al., 2022*) exhibits the highest effect capability in terms of loss rate, the model presented in this article is compared to literature (*Mehbodniya et al., 2022*) for diversified display capability. The experimental training sets are categorized into three groups based on information metrics a, b, and c, respectively, and input into the built training model. It should be noted that the interconnection coefficient g represents the physical variable related to low-carbon information, and the calculation result of the interconnection coefficient g will vary with the change of the training data input. $g = D \cdot \frac{\eta}{u}$, $\eta (\eta \in \{a, b, c\})$ is the green low carbon information to be displayed after performing clustering, and $u$ represents the constant measurement coefficient.

This article includes a similarity-based dimension alignment algorithm, which effectively selects relevant dimensions for data mining and visualization display, thereby reducing the interference of irrelevant dimensions during mining and visualization tasks.

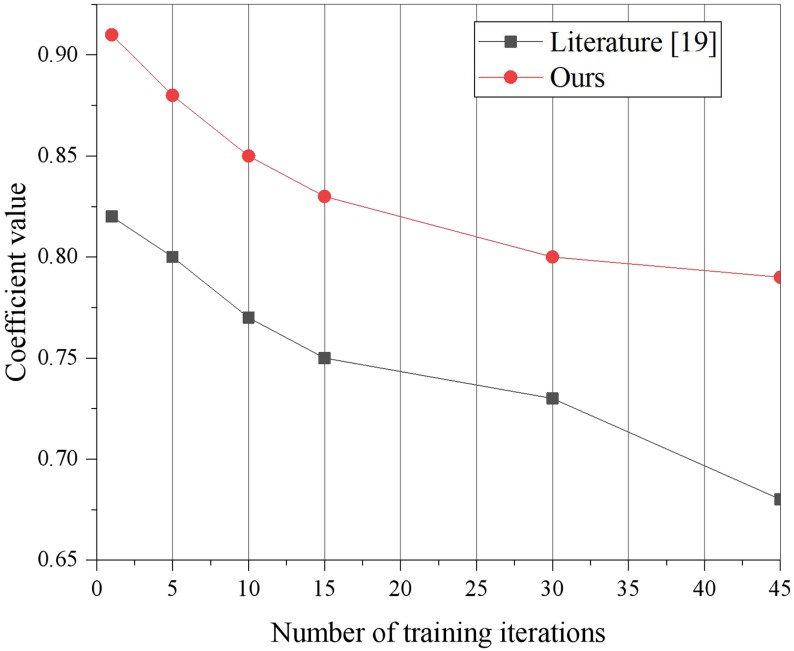

**Figure 7  Measure coefficient change curve contrast.**

Additionally, the proposed scheme performs correlation analysis on attributes, utilizing feature cleaning to filter out statistically irrelevant or weakly correlated attributes, while retaining the most pertinent attributes for mining or display purposes. As a result, the interconnection coefficient of the model presented in this article is consistently higher than that of literature (*Mehbodniya et al., 2022*) for different numbers of training iterations.

To demonstrate the superiority of the proposed model and provide a comprehensive evaluation, we conducted a thorough analysis with various experiments and data processing steps. Firstly, we applied a loss function analysis on both the Python collection and the dataset obtained from the DataFountain platform. This allowed us to evaluate the model's performance on diverse datasets and determine its generalizability. Figure 8 provides a visual representation of the results obtained from this analysis, depicting the model's performance and loss values.

Before proceeding with the analysis, we preprocessed the overall data to ensure its quality and consistency. We performed data cleaning, normalization, and feature extraction to eliminate noise and enhance the dataset's representativeness. Subsequently, we focused on the "green low carbon" information from 2000 to 2015, a critical period marked by the evolution of multimedia technology and increasing emphasis on environmental concerns. For training and data mining, we carefully selected the relevant data, ensuring that it accurately represents the characteristics of the green low-carbon domain. This meticulous data selection process was crucial to training the model effectively and fostering its understanding of the complex relationships within the green low-carbon information.

During the training process, we monitored the change in loss values closely. The progressive reduction in loss values is a strong indicator of the model's ability to learn and

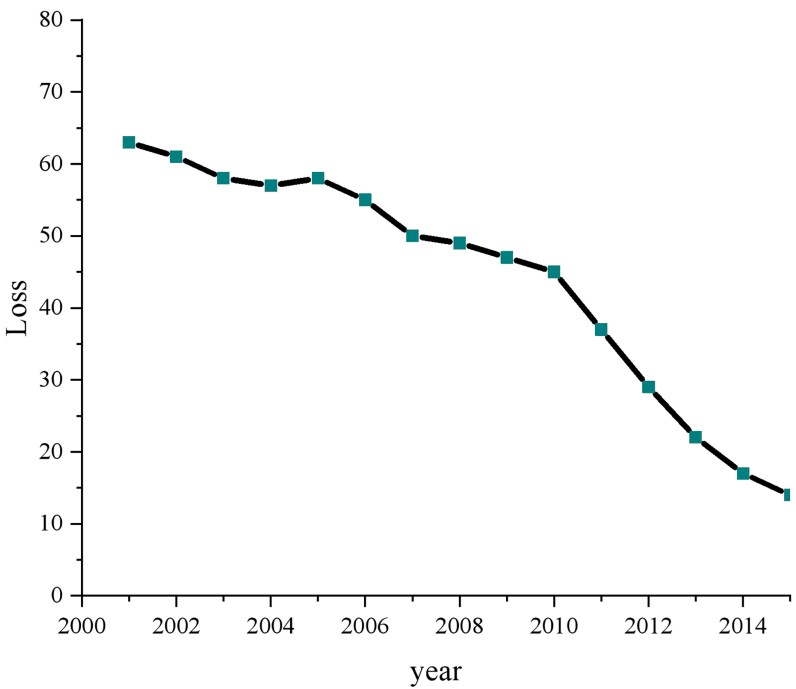

**Figure 8 Compared with the operation efficiency of other schemes.**

adapt to the given data. By utilizing multi-dimensional data processing and mining clustering techniques, the proposed model effectively exploited the intrinsic patterns and structures present in the data, leading to significant loss reduction. This demonstrated the model's capacity to process and comprehend complex information, which is pivotal in achieving more effective control over the green low-carbon domain.

## Discussion

The rapid advancement of information technology and network infrastructure has resulted in an exponential surge in data volume, particularly driven by the proliferation of multimedia content. Effectively processing and transforming this data into meaningful insights has emerged as a pivotal concern. To tackle the challenge of visualizing eco-friendly and low-carbon information within the context of urban environment design, this article employs cutting-edge data mining technology to construct a framework for knowledge extraction from information. Additionally, a similarity-based dimensional alignment algorithm is employed to process multidimensional data, discerning feature data that necessitates visualization display. Subsequently, a sophisticated visualization platform is meticulously constructed, employing Vue technology.

An intriguing finding from our analysis was the correlation between advancements in multimedia technology and the comprehensiveness of open green low-carbon information. The visualization in Fig. 8 vividly showcased the increasing completeness of open green low-carbon data over recent years. This observation highlighted the model's

adaptability to evolving data landscapes and its potential to contribute to real-world applications in green technology and environmental sustainability.

Moreover, we focused on enhancing the overall visualization structure of the model to improve its reliability and accuracy. A more interpretable and intuitive visualization not only helps researchers better understand the model's inner workings but also boosts the model's practical applicability. Users can confidently trust the model's outputs and utilize them in decision-making processes, making it a valuable asset in various green technology domains.

The analysis of experimental results reveals that the proposed scheme demonstrates a remarkable enhancement in data information mining, thereby augmenting the availability of eco-friendly low-carbon information. This approach adeptly filters out redundant information, consequently minimizing the loss rate in model training. Moreover, it exhibits robust data diversification display capabilities, thereby effectively propelling the realization of sustainable and high-quality development. It further promotes the application of advanced green technology, bolsters eco-friendly low-carbon management capability, accelerates network green upgrade, and elevates the level of eco-friendly power usage. The implementation of this model fosters innovation-driven quality transformation, power utilization, and efficiency enhancement, thus facilitating the realization of sustainable and high-quality development within the industry.

In conclusion, the extensive analysis and experiments conducted in this study affirm the superiority of the proposed model. The incorporation of diverse datasets, rigorous data processing, and the use of advanced techniques have collectively contributed to the model's exceptional performance. By efficiently reducing loss values, adapting to dynamic data trends, and providing reliable visualizations, the model exhibits significant potential for practical applications in the field of green low-carbon technology and beyond. Our research serves as a stepping stone for further advancements in this domain and offers valuable insights for researchers, policymakers, and industry professionals.

## CONCLUSION

In recent years, the combination of visualization and data mining methods has gradually demonstrated its powerful data analysis capabilities and interactive efficiency, thus providing a promising solution to the visualization of green and low-carbon information in urban environment design in the context of multimedia. To this end, this article proposes an enhanced data mining technology that constructs a knowledge extraction framework for green and low-carbon information. The similarity-based dimensional arrangement algorithm is also improved to effectively remove redundant or bad data, enhancing the visualization effect of green and low-carbon multidimensional information. By utilizing Vue for data visualization display, this article realizes direct forwarding and feedback of data and views, thereby empowering the entire society to reduce carbon emissions and promote peak performance. Moreover, the digital and green transformation of information in urban environmental design is facilitated, accelerating the service supply capability of deep integration of digital technology and vertical industry applications, and supporting the development of economic and social green transformation. Meanwhile,

Green low-carbon information contains multiple dimensions. In the visualization process, it is a challenge to choose which dimensions, the weights of the dimensions and how to display this information. Trade-offs and prioritization must be made, taking into account users' needs and goals, and minimizing redundancy and duplication of information.

In the longer term, the experts in the field of visualization at the IEEE VIS 2014 Workshop on Visualization in Predictive Analysis have presented a series of significant challenges in this direction, including training data and test data selection, feature selection and testing, model comparison and selection, model validation, model prediction quality control, and visualization-assisted prediction of complex problems, among others. In the future, we will continue to explore this research program and advance the process of information visualization in urban environment design.

## ACKNOWLEDGEMENTS
I thank the anonymous reviewers whose comments and suggestions helped to improve the manuscript.

### Funding
The author received no funding for this work.

### Competing Interests
The author declares that they have no competing interests.

### Author Contributions
- Junru Wang conceived and designed the experiments, performed the experiments, analyzed the data, performed the computation work, prepared figures and/or tables, authored or reviewed drafts of the article, and approved the final draft.

### Data Availability
The raw data is available at Zenodo: Guttikunda, Sarath K, & KA, Nishadh. (2022). Evolution of India's PM2.5 Pollution Between 1998 and 2020 Using Global Reanalysis Fields [Dataset]. Zenodo. https://doi.org/10.5281/zenodo.7115052.

### Supplemental Information
Supplemental information for this article can be found online at http://dx.doi.org/10.7717/peerj-cs.1614#supplemental-information.

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
