# Peer review of "Visual design of green information in urban environment based on global similarity calculation and multi-dimensional visualization technology"

_PeerJ Computer Science, doi:10.7717/peerj-cs.1614_

## Round 0.1 · original submission · Major Revisions

Dear authors

Thanks for your submission, your article has been reviewed by the experts in the field and you will see that they have some valid comments for the improvements of the article, therefore you are requested to please improve in the light of these comments.

Please also improve the language of the manuscript professionally,

Also, improve the abstract to make it a reasonable look to the readership.

**Language Note:** The Academic Editor has identified that the English language must be improved. PeerJ can provide language editing services - please contact us at copyediting@peerj.com for pricing (be sure to provide your manuscript number and title). Alternatively, you should make your own arrangements to improve the language quality and provide details in your response letter. – PeerJ Staff

Reviewer 1 ·

Basic reporting

In this paper, the framework of knowledge extraction is constructed by combining data mining and visualization, and general data mining is realized by using multidimensional visualization technology for visualization enhancement. At the same time, the dimension control and display of data visualization are carried out for green and low-carbon application scenarios in urban environment design, and the order of data dimensions is planned by using a dimension similarity algorithm. The experimental results verify that the above algorithm improves the visualization effect and effectively promotes deep cooperation between the industrial chain and supply chain.

Experimental design

Provide a detailed analysis of the experimental results, highlighting the performance metrics achieved by the proposed algorithm. Include statistical significance tests and comparative analysis against existing methods, if applicable.

Validity of the findings

1. Provide a brief literature review: Introduce existing research and technologies related to visual displays, data mining, and multidimensional visualization in urban environmental design. This will establish the current state of the field and highlight the novelty of the proposed approach.

2. Provide a clear explanation of the global similarity algorithm and its relevance to calculating attribute component quantities in specific dimensions of the vector. This will help readers understand the computational aspects of the proposed approach.

3. How do they enable two-way interaction and enhance visual displays? Please provide more details on the logical relationship and its significance.

Additional comments

1. It is noted that your manuscript needs careful editing by someone with expertise in technical English editing paying particular attention to English grammar, spelling, and sentence structure so that the goals and results of the study are clear to the reader.

2. Discuss potential avenues for future research and how the proposed approach can be further improved or extended.

3. Highlight how the proposed approach advances state-of-the-art visual displays and data mining for green and low-carbon scenarios in urban environmental design.

4. Improve clarity and organization: Ensure that the abstract is well-structured and flows logically. Use clear and concise language to convey the key concepts and findings. Consider rephrasing complex sentences to enhance readability.

Reviewer 2 ·

Basic reporting

This paper improves the data mining technology, constructs the knowledge extraction framework on green and low carbon information, and improves the dimensional arrangement algorithm based on similarity, removes redundant or bad data, and improves the visualization effect of green and low carbon multi-dimensional information.
By addressing the following suggestions, the revised paper will provide a clearer understanding of the research objectives, methodologies, experimental results, and contributions, thereby improving its overall quality and impact in the field of computer science.

1. Clarify the specific problem statement: The abstract should clearly state the problem being addressed, such as the specific challenges in visualizing green and low-carbon information in urban environmental design.
2. Please elaborate on the memory-based clustering algorithm.
3. Describe the algorithm in more detail, including its underlying principles, assumptions, and how it specifically addresses the challenge of information extraction for visual displays.
4. Clarify the role of Vue and MVVM pattern: Explain the specific functionalities and benefits of using Vue and the MVVM pattern in the proposed system.
5. Describe the experimental setup, datasets used, and evaluation metrics employed to validate the proposed algorithm. This will allow readers to assess the reliability and generalizability of the results.
6. Discuss limitations and potential extensions: Address any limitations or assumptions made in the study.
7. Clearly state the contributions: Explicitly mention the contributions of the paper in terms of novel algorithms, methodologies, or technological advancements.

Experimental design

.

Validity of the findings

.

---

## Round 0.2 · accepted · Accept

I am pleased to inform you that the revised version of the paper is acceptable and recommended by the experts. Therefore congratulations and thank you for your fine contribution

Reviewer 1 ·

Basic reporting

All the sections are revised according to previous comments.

Experimental design

The experimental section is revised according to suggestions.

Validity of the findings

This manuscript will be valuable for the researchers.

Reviewer 2 ·

Basic reporting

The article is revised as per the comments and I believe it’s suitable for publication in this current form.

Experimental design

The article is revised as per the comments and I believe it’s suitable for publication in this current form.

Validity of the findings

The article is revised as per the comments and I believe it’s suitable for publication in this current form.

Additional comments

The article is revised as per the comments and I believe it’s suitable for publication in this current form.